

# *PTMphinder*: an R package for PTM site localization and motif extraction from proteomic datasets

Jacob M. Wozniak[1,2] and David J. Gonzalez[1,2]

[1] Department of Pharmacology, University of California, San Diego, La Jolla, CA, United States of America
[2] Skaggs School of Pharmacy and Pharmaceutical Sciences, University of California, San Diego, La Jolla, CA, United States of America

## ABSTRACT

**Background**. Mass-spectrometry-based proteomics is a prominent field of study that allows for the unbiased quantification of thousands of proteins from a particular sample. A key advantage of these techniques is the ability to detect protein post-translational modifications (PTMs) and localize them to specific amino acid residues. These approaches have led to many significant findings in a wide range of biological disciplines, from developmental biology to cancer and infectious diseases. However, there is a current lack of tools available to connect raw PTM site information to biologically meaningful results in a high-throughput manner. Furthermore, many of the available tools require significant programming knowledge to implement.

**Results**. The R package *PTMphinder* was designed to enable researchers, particularly those with minimal programming background, to thoroughly analyze PTMs in proteomic data sets. The package contains three functions: parseDB, phindPTMs and extractBackground. Together, these functions allow users to reformat proteome databases for easier analysis, localize PTMs within full proteins, extract motifs surrounding the identified sites and create proteome-specific motif backgrounds for statistical purposes. Beta-testing of this R package has demonstrated its simplicity and ease of integration with existing tools.

**Conclusion**. *PTMphinder* empowers researchers to fully analyze and interpret PTMs derived from proteomic data. This package is simple enough for researchers with limited programming experience to understand and implement. The data produced from this package can inform subsequent research by itself and also be used in conjunction with other tools, such as motif-x, for further analysis.

Corresponding author
David J. Gonzalez,
djgonzalez@ucsd.edu

## INTRODUCTION

The human genome contains tens of thousands of genes that encode for proteins, the functional unit of the cell. Protein activity can be modulated by many factors such as expression level, alternative-splicing, localization and post-translational modifications (PTMs). Common PTMs range from small molecules (phosphorylation (*Hunter, 1995*; *Cohen, 2002*) and acetylation (*Kouzarides, 2000*; *Verdin & Ott, 2015*)) to peptide features

(ubiquitination (*Hicke, 2001*; *Hicke & Dunn, 2003*)) to sugars moieties (glycosylation *Dennis, Granovsky & Warren, 1999*; *Lis & Sharon, 1993*).

When PTMs are taken into consideration, the complexity of the proteome exponentially increases to over one million predicted protein products. Consequently, the analysis of protein PTMs has led to the deeper understanding of biological systems and development of therapies for a wide-range of maladies from cancer (*Shiloh, 2003*; *Sebolt-Leopold & Herrera, 2004*; *Cohen, 2001*) to infectious diseases (*Lapek Jr et al., 2017*; *Schmutz et al., 2013*) to auto-immune disorders (*Cohen, 2001*; *Mustelin, Vang & Bottini, 2005*). Thus, assessing the PTM status of proteins under various physiological conditions and disease states has the potential to facilitate new biological discoveries.

While many studies interrogate the PTM status of a single or a few proteins via conventional techniques (i.e., western blot), PTMs are rarely singular events and a simple stimulus can induce a massive number of alterations in protein modifications. In addition, conventional studies are limited by a lack of reliable antibodies for specific PTM site detection. To address these gaps, mass spectrometry-based proteomics has emerged as a powerful tool to simultaneously analyze the PTM status of thousands of proteins at tens of thousands of sites (*Macek, Mann & Olsen, 2009*; *Riley & Coon, 2016*). These approaches have led to many discoveries, from complicated signaling networks induced during infection (*Lapek Jr et al., 2017*; *Schmutz et al., 2013*) to the identification of motifs recognized by particular kinases (*Schwartz & Gygi, 2005*; *Chou & Schwartz, 2011*; *Wagih et al., 2016*). With the further development of such technology, one can expect the depth of proteomic-based PTM analyses to increase substantially.

To date, there are a number of tools available that allow users to analyze PTMs within proteomic data (*Chou & Schwartz, 2011*; *He et al., 2011*; *Ritz et al., 2009*). A common goal of these programs is to extract statistically-enriched motifs surrounding modification events for subsequent interpretation. While useful, many of these applications require significant preprocessing of raw data and substantial programming knowledge to implement. Furthermore, most of the available tools only support limited model organisms (*Chou & Schwartz, 2011*; *He et al., 2011*), forcing users to generate their own motif backgrounds for statistical fidelity. Thus, for biologists with limited programming experience or investigating less-studied organisms, a gap in expertise and software exists to fully analyze the PTMs within proteomic data sets.

Here, we introduce the R package *PTMphinder* for the analysis of PTMs identified in proteomic data in an effort to facilitate biological discoveries. The primary function of *PTMphinder*, phindPTMs, allows users to localize PTMs, detected as modified peptide residues in shotgun proteomic experiments, to their precise location in the full-length protein and extract neighboring motifs. In addition, *PTMphinder* contains auxiliary functions, parseDB and extractBackground, that reformat protein databases for downstream analyses and extract proteome-specific motif backgrounds, respectively. Collectively, these functions allow researchers to delve deeper into the PTMs identified within their proteomic data, enabling high-throughput analyses and new biological discoveries. *PTMphinder* is an open source package freely available for download from GitHub ('jmwozniak/PTMphinder").

## MATERIALS & METHODS

### Data sources

The example Human proteome used in this study was downloaded from Uniprot on 12/12/18 (https://www.uniprot.org/proteomes/UP000005640). The example shotgun proteomics data used in this study was published previously (*Lapek Jr et al., 2017*). The rmotix package (*Wagih et al., 2016*) was downloaded from GitHub (https://github.com/omarwagih/rmotifx).

### PTM localization methodology

All software was written in the R programming language. The algorithm for PTM localization and motif extraction receives the experimentally determined peptide sequence, total number of modifications, and potential PTM locations and scores as input. The program then extracts the represented proteins from a parsed proteome database, scans the full protein sequences for the detected peptides, and outputs the location in the full protein along with the flanking sequences. The extracted data is formatted into a new data table that can be exported or analyzed further within the R framework.

### PTM localization validation

For initial validation of the package, a pseudo-dataset (Table S1) was created using Microsoft Excel to match the format of the phindPTMs function input. Second, a previously published phospho-proteomic dataset was used to validate the use of the package on real-world data. The second dataset contained PTM information in the form of ptmRS output, which has to be modified prior to input into phindPTMs. The data was reformatted to match the phindPTMs function input using the text-to-column and find/replace functions in Microsoft Excel. This reformatted input table is provided as an example file with the *PTMphinder* package (phindPTMs_Input_Example.csv).

### Assessment of time of motif extraction

The time of motif extraction was measured for manual extraction and using *PTMphinder* with a previously published data set (*Lapek Jr et al., 2017*). First, the time of extraction per peptide was measured when manually extracting flanking sequences for five randomly selected peptides using standard spreadsheet and text editor software. Next, the time of extraction per peptide was measured when using *PTMphinder* within R. The starting point for each method was the same input file (phindPTMs_Input_Example.csv) and protein database (Human_Uniprot_Parsed_Example.txt).

### Code availability

An open source version of *PTMphinder* is freely available for download from GitHub (https://github.com/jmwozniak/PTMphinder).

## RESULTS AND DISCUSSION

The *PTMphinder* package contains three functions: parseDB, phindPTMs and extractBackground (Table 1) to assist in the analysis of PTMs contained within proteomic data sets.

**Table 1** *PTMphinder* functions and data.

| Function/Data | Description |
|---|---|
| parseDB | parses proteome database into a table with two columns: protein accession ID and sequence |
| phindPTMs | localizes PTM sites in full length proteins and extracts surrounding motifs |
| extractBackground | extracts motif backgrounds from a parsed proteome database |
| Human_Uniprot_Example.txt | Uniprot database example for input into parseDB |
| Human_Uniprot_Parsed_Example.txt | parsed Uniprot database example for input into phindPTMs and extractBackground |
| phindPTMs_Input_Example.csv | phospho-proteomic dataset example for input into phindPTMs |

## Format a uniprot database

The parseDB function reformats a proteome database for subsequent input into the phindPTMs and extractBackground functions (Fig. 1). This function extracts the protein ID and full protein sequence from the provided database and creates a data table with those two columns, respectively. The resultant data table can be further analyzed with R using various functions or written to a file and opened in a text or spreadsheet editor. This function is capable of handling two of the most common proteome databases used, Uniprot and RefSeq, as well as homemade databases provided the protein ID is the only value in ".fasta" description lines. Users can specify their database sources using the database source option ("db_source"). We recommend that users remove any redundant, reverse or contaminant sequences from the database prior to parsing. However, for convenience, we have included an option to remove any entries that start with "reverse" or "contaminant" if the filter option ("filt") is set to TRUE. We have provided both an unformatted and parsed Human Uniprot database as examples (Table 1).

## Localize PTM sites and extract motifs

Many proteomic workflows export PTM data that is difficult to manipulate and format properly for subsequent analysis. The phindPTMs function reads in experimentally acquired proteomic data, as well as a parsed reference database, to localize the detected PTMs in their respective proteins and extract relevant motifs. Users should re-format the proteome database used to search the mass spectrometry data with the above parseDB function prior to running phindPTMs. The experimentally acquired proteomic data must have the following six columns (described in Table 2): identifier, Protein_ID, Peptide_Seq, Total_Sites, PTM_Loc, and PTM_Score. Column headers need to be spelled exactly as above for proper import of data. For peptides with multiple modifications, the PTM_Locs and PTM_Scores should be separated by a semi-colon (";"). phindPTMs returns a data table with the following eight columns (described in Table 3): identifier, Protein_ID, Pep_Loc, Prot_Loc, Score, Flank_Seq, Ambiguity, Prot_Seq. The location of the PTMs in the full protein (Prot_Loc) can be further analyzed in the context of previous knowledge regarding that site using tools specific to the PTM of interest, such as PhosphoSitePlus (*Hornbeck*
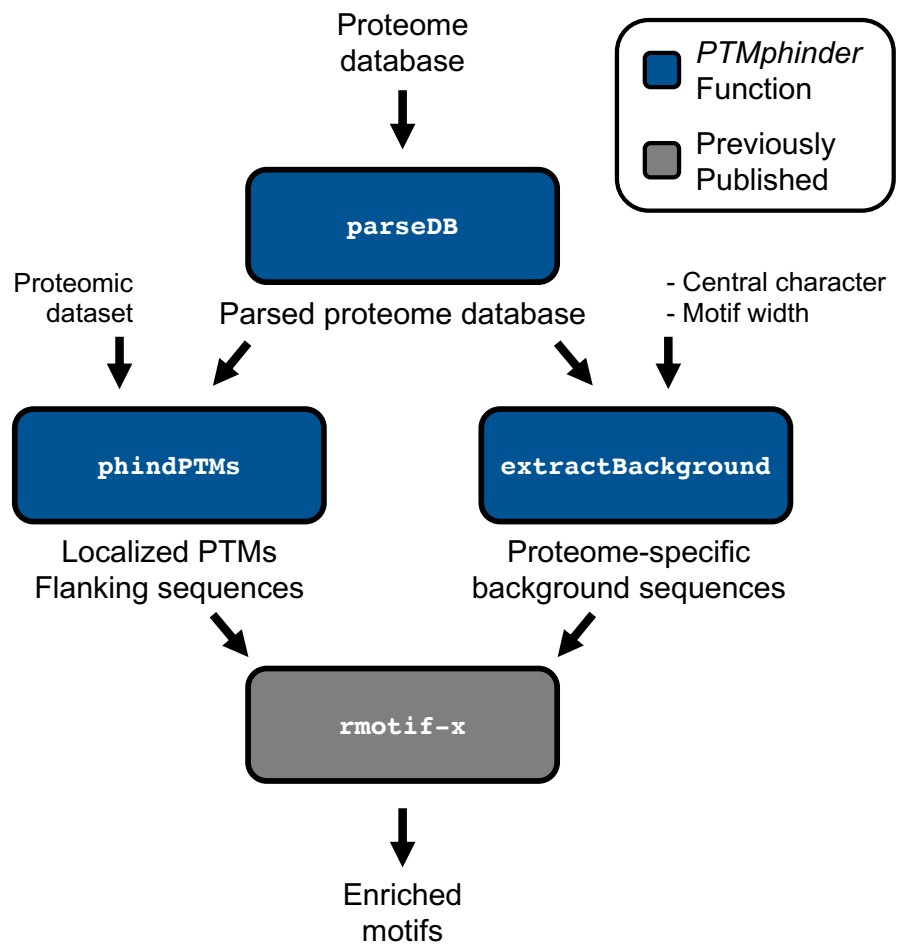

**Figure 1** **Workflow of *PTMphinder* Package.** Blue boxes indicate *PTMphinder* functions while grey bocks indicate functions outside of *PTMphinder*. Pipeline simply requires the user's proteomic dataset, a proteome database and minor, additional user-defined parameters.

**Table 2** **phindPTMs input columns.**

| Column Name | Description |
| --- | --- |
| Identifier | unique identifier for each modified peptide |
| Protein_ID | protein accession ID |
| Peptide_Seq | peptide sequence detected in experiment |
| Total_Sites | total number of modified sites on peptide |
| PTM_Loc | (potential) locations of PTM sites |
| PTM_Score | confidence scores of PTM localizations |

*et al., 2015*) or KinaseNet (http://www.kinasenet.ca/) in the case of phosphorylation events. The flanking sequences of each modified site (Flank_Seq) can be used as input into the a motif-finding tool (such as motif-x, see below) for subsequent motif enrichment analysis.

**Table 3  phindPTMs output columns.**

| Column name | Description |
| --- | --- |
| Identifier | unique identifier for each modified peptide |
| Protein_ID | protein accession ID |
| Pep_Loc | location of the PTMs in the identified peptide |
| Prot_Loc | location of the PTMs the full-length protein |
| PTM_Score | confidence scores of PTM localizations |
| Flank_Seq | flanking sequences extracted from PTMs |
| Ambiguity | ambiguity of PTMs based on input data |
| Prot_Seq | full-length protein from which motifs were extracted |

## Validation of the phindPTMs function

To validate the results of the phindPTMs function, two approaches were undertaken. First, a hypothetical pseudo-dataset was generated, which contained a variety of PTMs and scores that resemble the output from common proteomics software. This data set was input into phindPTMs function and the results were manually verified. We found that phindPTMs correctly localized and extracted the flanking sequences of 100% of the modified sites contained within the pseudo-dataset. Next, a real-world, phospho-proteomic data set was analyzed using the phindPTMs function. Again, manual validation found that the software correctly localized and extracted the flanking sequences of 100% of the modified sites. These results demonstrate that users can confidently trust the output of the package using data containing various PTMs and from real-world experiments.

## Extract proteome-specific background motifs

In order to properly perform motif enrichment analyses, a set of background motifs needs to be provided for statistical reasons. The motifs around every occurrence of a particular amino acid are commonly used as background for this purpose. However, many motif tools have limited proteome backgrounds to choose from (*Chou & Schwartz, 2011*; *He et al., 2011*) or simply use 10,000 random sites (*Wagih et al., 2016*). Using another species background or a small background set may result in abberent enrichment results. By using the extractBackground function contained in this package, users can generate their own background datasets from any proteome database. Furthermore, this function gives the user the flexibility to specify the width and central character of the background motifs. This tool simply reads in a parsed proteome database (from the parseDB function above), the central character and the width of the desired motifs. The output can be directly input into a motif enrichment tool, such as motif-x, as a list of background sequences (see below).

## Integration with existing tools

As with any new software, intergration with existing tools drastically increases its applicability to a variety of problems and accessibility to different users. Therefore, we designed the output of our tool to interface directly with existing tools for motif enrichment analysis, such as motif-x. With minimal reformatting, a user can simply use the output from the phindPTMs and extractBackground functions as input into motif-x
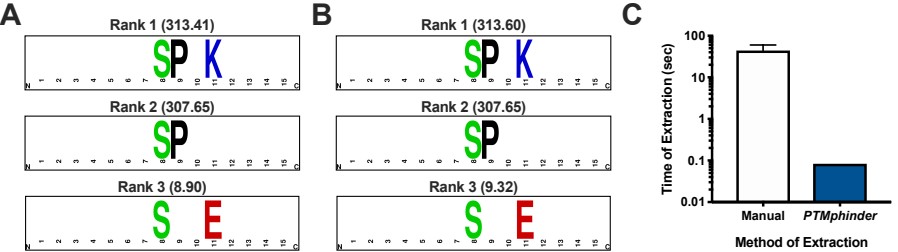

**Figure 2** **Assessment of *PTMphinder* functionality.** (A) Results of running rmotif-x on flanking peptide sequences already extracted in a previous publication[12]. (B) Results of running rmotif-x on flanking peptide sequences extracted using *PTMphinder* from the raw phospho-proteomics data in the same publication. Motif enrichment scores are represented in the parentheses. Depictions of motifs were generated using WebLogo (https://weblogo.berkeley.edu/). (C) Time of motif extraction for various methods ($n = 5$ modified peptides).

(Fig. 1). Fully-documented code to implement this workflow with the example files is provided as an "'.R'' script (File S1). This example uses real-world phospho-proteomic data previously published from our lab (*Lapek Jr et al., 2017*). Through implementing this package, a user can replicate the results reported in the previous publication in a simpler, more high-throughput manner (Figs. 2A, 2B). The results are nearly identical except for minor score differences due to an update in the Human Uniprot database since the initial publication of the data.

## Comparison with existing tools

While there is no open-source tool to our knowledge that performs the functions of *PTMphinder*, we compared the efficiency of *PTMphinder* to the manual extraction of motifs. We found that using *PTMphinder* resulted in a decrease in motif extraction time of over two orders of magnitude compared to manually extracting the sequences (Fig. 2C). Additionally, *PTMphinder* provides the location of the modified site in the full-length protein and reduces the potential of human error, effectively enhancing research reproducibility.

## Limitations and future directions

The current version of this package is limited in the input and output data formats; the input needs to resemble the example data (phindPTMs_Input_Example.csv), which requires some initial pre-processing, and the output has only been tested for compatibility with motif-x. Expanding this package to include other input/output formats, such as mzTab, and developing a more user-friendly interface would enhance its accessibilty and value to the scientific community. We are planning on implementing these changes in future versions of the package. Furthermore, including the package with other proteomic tools on BioConductor (https://www.bioconductor.org/) could increase the discoverability of *PTMphinder*. We hope to submit the package to BioConductor following this initial publication.

## CONCLUSIONS

Overall, *PTMphinder* allows researchers with limited prior programming experience to explore their proteomic data more completely. In a high-throughput manner, users are able localize PTMs in full-length proteins and extract biologically relevant motifs. The ability to isolate this additional information from proteomic studies can have a significant impact on data interpretation and subsequent hypothesis formulation. We have provided the source code for this package as well as detailed instructions for its use. Further development of this package for use with additional data input/output formats will increase the usefulness and accessibility of this platform to a wide-range of biological researchers.

## ACKNOWLEDGEMENTS

The authors would like to acknowledge John D. Lapek Jr. for thoughtful discussions about the project, beta-testing the package and help with the naming of the package.

### Funding

Jacob M. Wozniak is supported by the UCSD Graduate Training Programs in Cellular and Molecular Pharmacology and Rheumatic Diseases Research through institutional training grants from the National Institute of General Medical Sciences (T32 GM007752) and the National Institute of Arthritis and Musculoskeletal and Skin Diseases (T32 AR064194). David J. Gonzalez is supported by the Ray Thomas Edwards Foundation and the UC President's Office. The funders had no role in study design, data collection and analysis, decision to publish, or preparation of the manuscript.

### Grant Disclosures

The following grant information was disclosed by the authors:
National Institute of General Medical Sciences: T32 GM007752.
National Institute of Arthritis and Musculoskeletal and Skin Diseases: T32 AR064194.
Ray Thomas Edwards Foundation.
UC President's Office.

### Competing Interests

The authors declare there are no competing interests.

### Author Contributions

- Jacob M. Wozniak conceived and designed the experiments, performed the experiments, analyzed the data, contributed reagents/materials/analysis tools, prepared figures and/or tables, authored or reviewed drafts of the paper, approved the final draft.
- David J. Gonzalez conceived and designed the experiments, contributed reagents/-materials/analysis tools, authored or reviewed drafts of the paper, approved the final draft.

## Data Availability

PTMphinder is an open source package freely available at GitHub:
https://github.com/jmwozniak/PTMphinder

## Supplemental Information

Supplemental information for this article can be found online at http://dx.doi.org/10.7717/peerj.7046#supplemental-information.

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
