# Peer review of "PTMphinder: an R package for PTM site localization and motif extraction from proteomic datasets"

_PeerJ, doi:10.7717/peerj.7046_

## Round 0.1 · accepted · Accept

I really hope you can take the advice of the reviewers on board and ensure the software is made available via BioConductor for use by colleagues. I agree with their comment that it will maximise its use and get greater traction for your group.

Reviewer 1 ·

Basic reporting

No addition comments.

Experimental design

Methods are now described, although this does reveal how trivial the software is.

It's good to see that support for other PTMs (beyond phosphorylation) has now been added.

A PSI standard (mzTab) is now mentioned as future work, though what role it would play is not specified.

The authors' argument that "we would like to finalize the publication of our tool prior to uploading it to BioConductor to ensure that it is thoroughly tested and optimized" is concerning as it implies the software may not be fit for purpose at the present time.

Validity of the findings

Basic validation has now been added.

Additional comments

I do not expect this to be a high impact paper. The methods are straightforward, the software is simple (169 lines of R code) and without incorporation into a package such as BioConductor it will not be widely used.

Having said all that, none of these issues appear to preclude this revised version of the paper from being accepted for publication in PeerJ.